# MicroRNA Profiling as a Predictive Indicator for Time to First Treatment in Chronic Lymphocytic Leukemia: Insights from the O-CLL1 Prospective Study

**DOI:** 10.3390/ncrna10050046

**Published:** 2024-08-23

**Authors:** Ennio Nano, Francesco Reggiani, Adriana Agnese Amaro, Paola Monti, Monica Colombo, Nadia Bertola, Fabiana Ferrero, Franco Fais, Antonella Bruzzese, Enrica Antonia Martino, Ernesto Vigna, Noemi Puccio, Mariaelena Pistoni, Federica Torricelli, Graziella D’Arrigo, Gianluigi Greco, Giovanni Tripepi, Carlo Adornetto, Massimo Gentile, Manlio Ferrarini, Massimo Negrini, Fortunato Morabito, Antonino Neri, Giovanna Cutrona

**Affiliations:** 1Molecular Pathology Unit, IRCCS Ospedale Policlinico San Martino, 16132 Genoa, Italy; ennio.nano@hsanmartino.it (E.N.); monica.colombo@hsanmartino.it (M.C.); nadia.bertola@hsanmartino.it (N.B.); fabiana.ferrero@edu.unige.it (F.F.); franco.fais@unige.it (F.F.); giovanna.cutrona@hsanmartino.it (G.C.); 2SSD Gene Expression Regulation, IRCCS Ospedale Policlinico San Martino, 16132 Genova, Italy; 3Mutagenesis and Cancer Prevention Unit, IRCCS Ospedale Policlinico San Martino, 16132 Genoa, Italy; paola.monti@hsanmartino.it; 4Department of Experimental Medicine, University of Genoa, 16132 Genoa, Italy; ferrarini.manlio@gmail.com; 5Hematology Unit, Department of Onco-Hematology, Azienda Ospedaliera Annunziata, 87100 Cosenza, Italy; antonella.bruzzese@gmail.com (A.B.); enricaantoniamartino@gmail.com (E.A.M.); ernesto.vigna@aocs.it (E.V.); massim.gentile@tiscali.it (M.G.); 6Laboratory of Translational Research, Azienda USL-IRCCS di Reggio Emilia, 42122 Reggio Emilia, Italy; noemi.puccio@ausl.re.it (N.P.); mariaelena.pistoni@ausl.re.it (M.P.); federica.torricelli@ausl.re.it (F.T.); 7Clinical and Experimental Medicine PhD Program, University of Modena and Reggio Emilia, 41121 Modena, Italy; 8Institute of Clinical Physiology (IFC-CNR), Section of Reggio Calabria, 89124 Reggio Calabria, Italy; graziella.darrigo@cnr.it (G.D.); giovanniluigi.tripepi@cnr.it (G.T.); 9Department of Mathematics and Computer Science, University of Calabria, 87100 Cosenza, Italy; gianluigi.greco@unical.it (G.G.); carlo.adornetto@unical.it (C.A.); 10Department of Pharmacy, Health and Nutritional Science, University of Calabria, 87036 Rende, Italy; 11Department of Translational Medicine, University of Ferrara, 44121 Ferrara, Italy; massimo.negrini@unife.it; 12Gruppo Amici Dell’Ematologia Foundation-GrADE, 42122 Reggio Emilia, Italy; 13Scientific Directorate, Azienda USL-IRCCS di Reggio Emilia, 42122 Reggio Emilia, Italy

**Keywords:** microRNA, prognosis, CLL, time to first treatment (TTFT), *IGVH* mutations, del11q, del17p, Beta-2-microglobulin (B2M), Rai stage, *NOTCH1*

## Abstract

A “watch and wait” strategy, delaying treatment until active disease manifests, is adopted for most CLL cases; however, prognostic models incorporating biomarkers have shown to be useful to predict treatment requirement. In our prospective O-CLL1 study including 224 patients, we investigated the predictive role of 513 microRNAs (miRNAs) on time to first treatment (TTFT). In the context of this study, six well-established variables (i.e., Rai stage, beta-2-microglobulin levels, *IGVH* mutational status, del11q, del17p, and *NOTCH1* mutations) maintained significant associations with TTFT in a basic multivariable model, collectively yielding a Harrell’s C-index of 75% and explaining 45.4% of the variance in the prediction of TTFT. Concerning miRNAs, 73 out of 513 were significantly associated with TTFT in a univariable model; of these, 16 retained an independent relationship with the outcome in a multivariable analysis. For 8 of these (i.e., miR-582-3p, miR-33a-3p, miR-516a-5p, miR-99a-5p, and miR-296-3p, miR-502-5p, miR-625-5p, and miR-29c-3p), a lower expression correlated with a shorter TTFT, whereas in the remaining eight (i.e., miR-150-5p, miR-148a-3p, miR-28-5p, miR-144-5p, miR-671-5p, miR-1-3p, miR-193a-3p, and miR-124-3p), the higher expression was associated with shorter TTFT. Integrating these miRNAs into the basic model significantly enhanced predictive accuracy, raising the Harrell’s C-index to 81.1% and the explained variation in TTFT to 63.3%. Moreover, the inclusion of the miRNA scores enhanced the integrated discrimination improvement (IDI) and the net reclassification index (NRI), underscoring the potential of miRNAs to refine CLL prognostic models and providing insights for clinical decision-making. In silico analyses on the differently expressed miRNAs revealed their potential regulatory functions of several pathways, including those involved in the therapeutic responses. To add a biological context to the clinical evidence, an miRNA–mRNA correlation analysis revealed at least one significant negative correlation between 15 of the identified miRNAs and a set of 50 artificial intelligence (AI)-selected genes, previously identified by us as relevant for TTFT prediction in the same cohort of CLL patients. In conclusion, the identification of specific miRNAs as predictors of TTFT holds promise for enhancing risk stratification in CLL to predict therapeutic needs. However, further validation studies and in-depth functional analyses are required to confirm the robustness of these observations and to facilitate their translation into meaningful clinical utility.

## 1. Introduction

Chronic lymphocytic leukemia (CLL) is a B-cell disorder characterized by the monoclonal accumulation of CD5/CD23-positive lymphocytes at multiple sites [1,2,3] and a clinical heterogeneity [4,5] which has been correlated with different cytogenetic and molecular features of leukemic cells [6,7,8,9].

Many CLL patients do not require immediate therapeutic intervention, and for the majority, therapy can be initiated after a variable period of disease progression. To avoid useless treatment and to limit treatment only to the progressing cases, a “watch and wait” strategy is generally adopted in which only patients who progress, according to defined clinical criteria, are treated [1]. Due to the aforementioned variability of CLL, prognostic studies aimed at defining the risk of disease progression, warranting therapeutic intervention, have a relevant clinical impact [10,11].

While single prognostic factors, although relevant, may have a limited prognostic significance, the evaluation of multiple biomarkers with varying prognostic power may lead to more definite prognostic models [12]. To this end, prognostic scores have been developed that incorporate different biomarkers into comprehensive models. The CLL-International Prognostic Index (CLL-IPI) has emerged as one of the well-established models, demonstrating its efficacy in predicting overall survival, as well as time to first treatment (TTFT) and progression-free survival prediction in the context of chemo-immunotherapy [13,14]. Furthermore, our group has documented the predictive value of CLL-IPI for TTFT in early-stage CLL. In the same setting of Binet stage A CLL, Condoluci et al. introduced the International Prognostic Score for Early CLL patients (IPS-E) [15]. Further validated by Smolej et al. [16], the IPS-E and its alternative version (AIPS-E) effectively predicted TTFT and therefore may provide valuable guidance for clinical decision-making in early-stage CLL patients. In our prospective O-CLL1 study, evaluating newly diagnosed Binet stage A patients, both AIPS-E and IPS-E accurately predicted the need for therapy in early-stage CLL patients [17].

The advent of next-generation sequencing (NGS) has facilitated the discovery of an unprecedented number of gene mutations with potential prognostic and predictive value [9]. A recent study found that mutations were detected in roughly 35% of CLL patients, with frequencies ranging from 2.3% to 9.8% of cases, and that *NOTCH1* mutations were the most common [18]. When the predictive impact of nine recurrently mutated genes was tested, eight of them (*BIRC3*, *EGR2*, *NFKBIE*, *NOTCH1*, *POT1*, SF3B1, *TP53*, and *XPO1*) were independently associated with shorter TTFT [18].

In recent years, the role of microRNAs (miRNAs) in CLL has garnered increasing attention (reviewed in [19,20,21]). MiRNAs are evolutionarily conserved, single-stranded non-coding RNA molecules that can redundantly and simultaneously regulate the expression of multiple genes. Acting post-transcriptionally, miRNAs bind to the messenger RNA of target genes, affecting translation or inducing degradation [22,23]. 

Several miRNAs have been described to be associated with CLL prognosis [24,25,26,27,28,29,30] and pathogenesis [31,32]. In addition, miRNAs may contribute to the deregulation of apoptosis [31,32,33], BCR signaling [34,35,36], or metabolism [37] in CLL cells, and treatment of CLL patients with chemotherapy or BCR inhibitors was shown to affect the expression of miRNAs involved in these processes [38,39,40]. However, the pathways regulated by most miRNAs remain to be elucidated.

MiRNA expression profiling studies have disclosed correlations between certain miRNA signatures and cytogenetic features and/or *IGHV* gene mutational status [30,41], which are recognized prognostic markers in CLL. Finally, certain miRNA signatures are associated with disease progression and outcome [24,30,34,42,43] or with the onset of Richter’s transformation [44,45,46], a lethal condition characterized by the development of aggressive lymphoma in CLL patients [1,47].

Specifically, miR-15a and miR-16-1, which are located on 13q14 [31], act as tumor suppressors and were the first miRNAs used to predict outcomes in CLL [24]. Consistent with the notion that miR-15 and miR-16 regulate cell apoptosis and proliferation [33,48], our group showed that transfection of miR-15 and miR-16 mimics into del(13)(q14) CLL cells significantly hindered their growth in NOD/Shi-scid, γ(c)(null) (NSG) mice, resulting in substantial tumor regression [49]. Moreover, our previous research revealed that CLL has a miRNA expression profile that closely resembles that of antigen-experienced B cells; some miRNAs in this profile are likely to influence disease progression, as suggested by their correlation with the clinical outcomes [30,50].

In the context of our prospective O-CLL1 study, we herein explored the predictive role of 513 miRNAs on TTFT by (i) incorporating significant miRNAs into a basic prognostic model including *IGHV* mutational status, del11q and del17p, beta-2-microglobulin (B2M), *NOTCH1* mutation, and Rai stage, which are known to predict TTFT in CLL [51,52,53,54], and (ii) analyzing their potential association with 50 genes involved in TTFT that we previously identified by an artificial intelligence (AI)-based model. Our approach identified 16 out of 513 miRNAs independently associated with TTFT. Moreover, enrichment analysis was performed to evaluate biological processes regulated by the selected miRNAs, while correlation analysis and multiple bioinformatics software were used to evaluate possible interactions between the 16 prioritized miRNAs and the 50 genes selected as a CLL TTFT predictive signature, defined by an AI model [54].

## 2. Results

### 2.1. TTFT Prediction by the Basic Model

The estimated median TTFT for the 224 CLL cases from our prospective O-CLL1 study was 105.2 months (95% CI, 94.2–116.1 months), with 4-year and 8-year cumulative first-treatment-free survival of 76.8% (95% CI, 71.1–82.5) and 57.4% (95% CI, 50.1–64.6), respectively (Figure 1). Only six out of nine univariable predictors (i.e., Rai stage, B2M, *IGVH* and *NOTCH1* mutational status, 11q and 17p deletions) remained significantly associated with TTFT when tested in a Cox multivariable model adopting a backward elimination strategy. Univariable and multivariable Cox analyses, focusing on established TTFT predictors, are detailed in Appendix A. The six variables together gave a prognostic value for TTFT (i.e., a Harrell’s C-index) of 75% and an explained variation of 45.4%.

### 2.2. TTFT Prediction by miRNAs

Of the 513 available miRNAs tested, 73 were found to be significantly associated with TTFT in univariable Cox regression analyses (all *p* ≤ 0.05) (Table 1). These 73 miRNAs were then simultaneously included in the same multivariable Cox regression model, and those significantly associated with TTFT were identified by a backward elimination strategy.

Sixteen miRNAs retained an independent association with TTFT (Table 2). For eight miRNAs (i.e., miR-582-3p, miR-33a-3p, miR-516a-5p, miR-99a-5p, and miR-296-3p, miR-502-5p, miR-625-5p, and miR-29c-3p), lower expression levels were associated with a higher likelihood of treatment (i.e., shorter TTFT). Conversely, for the remaining eight miRNAs (i.e., miR-150-5p, miR-148a-3p, miR-28-5p, miR-144-5p, miR-671-5p, miR-1-3p, miR-193a-3p, and miR-124-3p), a higher expression was associated with shorter TTFT.

Using these 16 miRNAs, a risk prediction score (ranging from 0 to 100%) was derived (Figure 2).

Inclusion of the miRNA scores into the basic model (i.e., a Cox model including Rai stage, B2M, *IGVH* and NOTCH1 mutational status, 11q and 17p deletions) significantly increased both the Harrell’s C-index (from 75.0% to 81.1%) and the explained variation in TTFT (from 45.4% to 63.3%). Remarkably, the inclusion of the miRNA scores into the basic model also yielded an IDI and an NRI of +14.9% and +44.2%, respectively (Table 3).

Appendix A illustrates the survival curve derived from the multivariable Cox proportional hazards model, which closely aligns with the Kaplan–Meier curve presented in Figure 1, thereby confirming the robustness and calibration of the model for predicting TTFT.

### 2.3. Correlation and Interaction Analysis between miRNAs and Genes Found to Be Related to TTFT by an AI Model

Next, we analyzed the relationship between the 16 miRNAs identified as predictive of TTFT and the previous 50 AI model prioritized genes (AI genes) that we found significantly associated with TTFT in the same patients of the O-CLL1 cohort [54]. Fifteen out of sixteen miRNAs showed at least one significant negative correlation with the AI genes; in particular, eight miRNAs (i.e., miR-29c-3p, miR-625-5p, miR-150-5p, miR-144-5p, miR-28-5p, and miR-516a-5p) showed a relationship with one of the top ten genes selected by the neural network (i.e., *CEACAM19*, *PIGP*, *FADD*, *FIBP*, *IGF1R*, *COL28A1*, *QTRT1*, *MKL1*, *GNE*, *SLC39A6*) (Figure 3). Only hsa-miR-124-3p showed no correlation with any of the AI genes; in addition, miR-29c-3p had a significant negative correlation with the highest number of genes prioritized by the AI model.

Comparing the previous correlation analysis with the miRComb model, this analysis retained only five miRNA–mRNA annotated interactions, displaying a mild to moderate correlation index. miR-29c-3p showed again the highest number of interactions, three of which were previously predicted; specifically, miR-29c-3p’s interaction with *PRICKLE1* was annotated in the microCosm database, while interactions with *ANKRD52* and *ZBTB34* were annotated in the TargetScan database. miR-29c-3p’s interaction with *KDM5B* was also predicted by TargetScan with experimental validation, although with less strong evidence. MiR-625-5p was found to be significantly anti-correlated with *IGF1R*, an interaction annotated by miRTarBase with a strong validation score (Appendix A). Interestingly, the MiRComb regulation score suggested an association between *IGHV* mutational status and the interactions between miR-29c-3p with the previously identified AI model-selected genes *KDM5B*, *ANKRD52*, *ZBTB34*, *PRICKLE1*; this dysregulation was not evident when considering the interaction between miR-625-5p and *IGF1R* (Figure 4). Although the magnitude of the effect of *IGHV* mutational status on the former interaction is strong, this is largely due to an over-expression of miR-29c-3p in the *IGHV* mutated group compared to the unmutated one and, to a lesser extent, to a slightly not significant downregulation of the previous four AI-selected genes in the *IGHV* mutated group compared to the *IGHV* unmutated one.

The aforementioned miRNA–mRNA interactions represented by the miRComb model were found to be specific for CLL. Although the statistical power might be affected by the small sample size, miRNA and mRNA expression profiles of tonsil B cells from normal individuals did not reveal any significant interactions between any of the 16 miRNAs and 50 AI-selected genes.

### 2.4. Pathways Regulation by the 16 miRNAs Linked to TTFT

To further investigate the biological significance of the 16 identified miRNAs, an enrichment analysis was carried out, showing that selected miRNAs are involved in cancer and cellular processes associated with disease progression and drug resistance.

Specifically, Kegg term overrepresentation analysis revealed a significant pathway (i.e., hsa05206) corresponding to the “MicroRNAs in cancer” pathway, associated with the miRNA genes MIR1-2, MIR28, MIR29C, MIR99A, MIR124-3, MIR150, and MIR625 encoding for miR-1-3p, miR-28-5p, miR-29c-3p, miR-99a-5p, miR-124-3p, and miR-150-5p, respectively (Table 4).

Overrepresentation analysis using Wikipath revealed six significant terms associated with the miRNA genes MIR29C, MIR33A, and MIR150 (Table 4). The terms WP1601, WP2249, and WP1545, signatures of “Fluoropyrimidine activity”, “Metastatic brain tumor”, and “miRNAs involved in DNA damage response”, respectively, were enriched in the MIR29C gene product; the terms WP299 and WP430, signatures of “Nuclear receptors in lipid metabolism and toxicity” and “Statin inhibition of cholesterol production”, respectively, were enriched in the MIR33A gene product. Finally, the WP2023 term, the signature of the “Cell differentiation expanded index”, was enriched in the MIR150 gene product.

## 3. Discussion

Predicting overall and progression-free survival in CLL is a dynamic field of research that adapts to advances in biology and therapeutics, providing insights to personalized patient planning and guiding clinicians in tailored treatment decisions. Conversely, models predicting TTFT, which is not influenced by the treatment choice, play a pivotal role in counseling, family planning, surveillance, and identification of high-risk candidates for potential early intervention in clinical trials [55,56]. Indeed, several predictive models have been published for this purpose [57,58,59]. Recently, the European Research Initiative on CLL (ERIC) introduced a score that categorizes patients based on *IGHV* mutation and somatic hypermutation status [18]. 

Nevertheless, most of these predictive models typically require a combination of clinically accessible factors and molecular factors, which are relatively straightforward to assess. In the O-CLL1 trial, multivariable analyses identified a foundational model comprising Rai stage, B2M, *IGHV*, and *NOTCH1* mutational status, in addition to 17p and 11q deletions [51,52,53,54]. This model also served as the basis for validating the improved prognostic efficacy of lymphocyte doubling time (LDT) [53] and, more recently, for incorporating our previously discovered AI gene expression data into an enhanced model predicting TTFT [54]. 

The current study focused on analyzing the profile of miRNAs at the time of diagnosis among treatment-naive CLL cases enrolled in the prospective O-CLL1 trial. The primary aim was to identify specific miRNAs that could function as predictive markers to better identify patients in need of early treatment. The establishment of a risk prediction score based on 16 miRNAs identified as independently associated with TTFT emphasizes the significant impact of miRNA data in predicting TTFT in CLL. Specifically, lower expression of eight miRNAs (i.e., miR-582-3p, miR-33a-3p, miR-516a-5p, miR-99a-5p, miR-296-3p, miR-502-5p, miR-625-5p, and miR-29c-3p) was associated with a shorter TTFT, while for the remaining eight miRNAs (i.e., miR-150-5p, miR-148a-3p, miR-28-5p, miR-144-5p, miR-671-5p, miR-1-3p, miR-193a-3p, and miR-124-3p), the higher expression was associated with a shorter TTFT.

Importantly, the integration of these miRNAs into the basic model significantly improved predictive accuracy, as reflected by the improved Harrell’s C-index from 75.0% to 81.1% and explained variation in TTFT from 45.4% to 63.3%. Additionally, this integration yielded an IDI of +14.9% and an NRI of +44.2%. This suggests that miRNA expression data may provide meaningful insights, offering a promising opportunity for counseling, and a more reasonable follow-up scheduling through early identification of cases with a higher likelihood of requiring early therapy.

To date, miR-15a and miR-16-1, localized on chromosome 13q14 and functioning as bona fide tumor suppressors [31,33,48], have been utilized as pioneering prognostic indicators in CLL [24]. Subsequent investigations have highlighted the potential of miRNA profiling in enhancing the accuracy of CLL prognostication. Specifically, the expression levels of dysregulated miR-155, miR-181b, miR-29a/b/c, and miR-34a have been systematically correlated with established prognostic biomarkers, including *IGHV* and *TP53* mutational status, as well as ZAP70 expression, thereby exerting a discernible impact on the clinical outcomes of CLL patients [25,28,41,43,60,61]. Indeed, our previous study unveiled several dysregulated miRNAs in CLL, indicating their potential role in the pathogenesis of the disease and their contribution to its progression, ultimately influencing the initiation of therapy [30]. In particular, 8 miRNAs (i.e., miR-146b-5p, miR-222-3p, miR-26a-5p, miR-29c-3p, miR-29c-5p, miR-502-3p, and miR-503-5p) out of 15 dysregulated miRNAs demonstrated a significant role in predicting TTFT, with miR-26a-5p and miR-532-3p remaining significantly associated with TTFT after data adjustment for confounders.

Notably, in our current study, only miR-29c-3p was found to be significantly associated with TTFT after applying the Cox multivariable regression model. However, it is essential to consider that, in contrast to our previous analyses and other published studies, our current investigation diverges from the conventional focus based on dysregulated miRNAs. Instead, we opted for a broader statistical strategy, encompassing the evaluation of all the assessable miRNAs to gain a more comprehensive understanding of the miRNA landscape in the context of TTFT prediction. In the present investigation, we have delineated a distinctive signature comprising 16 miRNAs that influence the risk of initiating therapy in CLL. Of note, elevated levels of eight of them correlated with higher treatment likelihood, suggesting their role in disease progression, while the remaining showed an inverse relationship, lowering treatment initiation likelihood, indicating their dual ability to act either as promoters of oncogenic processes or as regulators of tumor suppression.

We reasoned that the understanding of the regulatory networks involving the identified 16 miRNAs could be of interest to further elucidate the CLL pathogenesis; therefore, we adopted a comprehensive in silico approach employing correlation, interaction, and enrichment models to validate and refine miRNA–mRNA analyses in our O-CLL1 dataset. The miRNA–mRNA correlation analysis revealed at least one significant negative correlation between 15 of the identified miRNAs and the set of 50 AI model-based genes we previously identified, with miR-29c-3p being the one with the highest number of correlations as confirmed by multiple tools. miR-625-5p was found to be significantly anti-correlated only with *IGF1R*; interestingly, *IGF-1R* was functionally validated as a direct target of miR-625-5p also in melanoma cancer cells, being positively modulated by the long non-coding RNA LINC01291 via miR-625-5p sponging [62]. Of particular interest, miR-29c was previously found to exhibit differential expression based on the *IGHV* status [24,30], which correlated with the expression of some of its specific target genes [61,63]. The present study expanded this observation to four AI-generated genes (*ANKRD52*, *KDM5B*, *PRICKLE1*, and *ZBTB34*) previously described to be strongly associated with TTFT in CLL [54]. Regulation of *KDM5B* by miR-29c was described in endometrial carcinoma, showing a correlation between elevated levels of *KDM5B* and tumor grade and paclitaxel resistance [64]. Finally, the enrichment analysis revealed the multifaceted involvement of miRNAs in cellular activities, from tumor development and resistance to lipid metabolism regulation.

Our previous investigation demonstrated that IL-23R expression on CLL cells independently estimated TTFT in the O-CLL1 cohort, supporting the notion of an essential IL-23 autocrine loop driving CLL expansion [65]. Furthermore, we found that miR-146b-5p modulates IL-12Rβ1 expression, which influences TTFT, with lower levels associated with shorter TTFT duration [50]. Notably, in the current study, miR-148a-3p emerged as an independent prognostic factor in the multivariable model, showcasing a striking complementarity of 18 out of 21 bases with a sequence within the interleukin 23 subunit alpha (IL-23A) mRNA (miRWalk: refseq ID NM_016584, http://mirwalk.umm.uni-heidelberg.de/human/gene/51561/ accessed on 21 August 2024), thus expanding the exploration of IL23A gene regulation.

In conclusion, the identification of specific miRNAs as predictors of TTFT in CLL represents a promising avenue for refining risk stratification and predicting therapeutic needs. The integration of miRNA data into predictive models holds the potential to improve the accuracy of clinical decision-making in CLL management. However, further validation studies and comprehensive functional analyses are needed to confirm the reliability and robustness of these findings.

## 4. Materials and Methods

### 4.1. Patient Population and Study Design 

In the observational O-CLL1 study (clinicaltrials.gov identifier NCT00917540), 224 newly diagnosed Binet A CLL cases from 40 Italian institutions were prospectively enrolled for miRNAs analysis [30]. The miRNA expression data are deposited in the National Center for Biotechnology Information (NCBI) Gene Expression Omnibus repository (http://www.ncbi.nlm.nih.gov/geo/ accessed on 21 August 2024) and are accessible via GEO Series accession number GSE40533. The gene expression data are accessible through GEO Series accession number GSE40570.

All participants gave written informed consent, and the study was approved by the appropriate institutional review boards. Detailed inclusion and exclusion criteria have been previously outlined [54]. Specifically, the recruitment was limited to cases diagnosed within 12 months, age ≤ 70 years, and at Binet stage A [30].

### 4.2. Assessment of Biological Markers

The diagnosis was confirmed by flow cytometric analysis, which assessed the proportion of CD5/CD19/CD23 triple-positive B cells; monoclonal antibodies (mAbs) against CD19-FITC (BD Biosciences Pharmigen, San Jose, CA, USA), CD23-PE (BD Biosciences, San Jose, CA), and CD5-PC5 (Beckman Coulter Immunotech, Marseille, France) were used for this purpose. CD38-positive leukemic cells were quantified through triple staining with CD19 FITC (BD Biosciences), CD38 PE (BD Biosciences), and CD5 PC5 (Beckman Coulter Immunotech) mABs. CD38 positivity was defined with a cutoff of ≥20%, following previous reports [30].

ZAP-70 was detected by flow cytometry using a ZAP-70-FITC (clone 2F3.2, Millipore, Temecula, CA, USA) or an isotype control mAb (mouse IgG2a-FITC, BD Biosciences) as previously described [30]. Briefly, peripheral mononuclear cells purified from fresh heparinized CLL samples by Ficoll-Hypaque gradient were first incubated with CD3 PE-CY7, CD19 PE, and CD5 PC5 mAbs (BD Biosciences, San Jose, CA, USA), fixed, permeabilized with fix and perm reagents (Caltag Laboratories, Buckingham, UK), and exposed to ZAP-70 or the isotype control mAb. A cutoff value of more than 30% for ZAP-70 positivity was used, as previously reported [30] and calculated by receiver-operating characteristic analysis as the most suitable ZAP-70 cutoff value to discriminate *IGHV*-Unmutated (UM) from *IGHV*-Mutated (M) cases. All flow cytometric analyses were performed on a FACSCalibur flow cytometer (BD Biosciences).

Cytogenetic abnormalities, including chromosome deletions 11q23 and 17p13, were examined by fluorescence in situ hybridization (FISH) in a purified CD19-positive population, according to established protocols [30]. The mutational status of the *IGHV* gene was assessed on cDNA samples [30] by aligning sequences to the IMGT directory and by analyzing them using IMGT/V-QUEST software(version: 3.6.3 (30 January 2024). The *NOTCH1* mutation hotspot was determined by next-generation deep sequencing, as previously outlined [51].

### 4.3. miRNAs Analysis

Total RNA was extracted from CD19-positive purified B-cell samples using TRizol reagent (Life Technologies, Carlsbad, CA, USA). Subsequently, RNA quality was evaluated through the Agilent 2100 Bioanalyzer (Agilent Technologies). Any RNA sample with poor quality, as indicated by an RNA integrity number < 7, was excluded from microarray analyses. After sample collection, processing was performed according to the guidelines of the Agilent manual for the Human miRNA Microarray V2 platform (Agilent Technologies, Palo Alto, CA, USA), which includes miRNAs from the Sanger miRBase (v10.1). Expression values of miRNAs were computed using Agilent Feature Extraction Software 10.1, followed by summarization and background subtraction. MiRNAs with low expression (all detection calls missing), non-human miRNAs, and miRNAs expired according to Sanger miRBase release 15 (April 2010) were excluded. The obtained raw data underwent quantile normalization, conversion to positive values with a minimum value of 1, and log2 transformation using the R-2.14 statistical environment (http://www.r-project.org/ accessed on 21 August 2024) [30]. Array old identifiers were converted to the last version with Mirbase (i.e., miR-33a* = miR-33a-3p, miR-miR-99a = 99a-5p, miR-625 = miR-625-5p, miR-29c = miR-29c-3p, miR-150 = miR-150-5p, miR-148a = miR-148a-3p, miR-144* = miR-144-5p, miR-1 = miR-1-3p, and miR-124 = miR-124-3p).

### 4.4. miRNA–mRNA Correlation, Interaction, and Enrichment Analyses

To elucidate potential relationships between the miRNAs significantly associated with TTFT and the previously identified 50 genes implicated in TTFT as detected by an AI-based model [54], a Spearman correlation analysis between miRNA and mRNA expression profiles was performed using the O-CLL1 database [30]. To retain miRNA–mRNA correlations implicating an miRNA–mRNA interaction, an additional Spearman correlation analysis between expression profiles was performed using the miRComb model [66], selecting significant correlations that were suggestive of an interaction annotated in at least one of the following prediction databases, namely microCosm (version 5.18) and TargetScan 6.2. The sensitivity of the previous model was improved by extending the annotation query to the miRTarBase database [67], which annotated experimentally proven miRNA–mRNA interactions accompanied by a qualitative validation score including the two values “Low evidence” and “Strong evidence” based on the experimental techniques involved in the validation protocol. At the same time, the specificity of the model was increased by selecting correlations with a corresponding p-value, adjusted for multiple comparisons inherent to the whole gene expression and miRNA profile analysis, below a threshold of 0.05. Additionally, the mirComb model was applied to the expression dataset to calculate an miRNA–mRNA regulation score, which allowed the assessment of the effect of *IGHV* mutational status on each miRNA–mRNA interaction retained; the regulation score was calculated as follows:$$score=-2\left(\log_{2}ratio_{mRNA}\;\cdot \;\log_{2}ratio_{miRNA}\right)$$
where log_2_ *ratio_mRNA_* and log_2_ *ratio_miRNA_* equal to the log-transformed fold changes of mRNA and miRNA expressions of a given interaction pair, respectively, calculated by comparing the *IGHV* mutated group with the unmutated group. Specifically, the previous score measured the magnitude of interaction deregulation between an miRNA and an mRNA as a result of the effect mediated by a categorical variable: a positive score implied interaction deregulation, while a negative score indicated a preserved interaction. To ensure the specificity of the miRNA–mRNA interactions detected from the O-CLL1 cohort for CLL pathogenesis, a prior correlation analysis was also carried out on a control set of normal B-cell sample subpopulations collected from tonsils of healthy individuals. In particular, miRNA and mRNA expression profiles were collected for 3 samples of Naive B cells, 2 samples of Marginal Zone-like B cells, and 3 samples of memory cells (GSE51529).

To elucidate the biological significance of the identified miRNAs that are correlated with clinical outcomes, we conducted an over-representation analysis (ORA) of Kyoto Encyclopedia of Genes and Genomes (KEGG) and Wikipathways terms. This analysis was performed using the R package ClusterProfiler, a versatile tool designed for the interpretation of omics data [68]. ORA allows the identification of enriched biological pathways or processes associated with a given gene set, thereby providing insight into the potential functional roles of the miRNAs under investigation.

### 4.5. Statistical Analyses

TTFT was calculated during the watch-and-wait period, which lasted from the date of diagnosis to the start of therapy or the last follow-up. The prognostic impact of standard risk factors and miRNAs for TTFT was preliminarily investigated by univariable Cox regression analyses, with data presented as hazard ratios (HRs) with 95% confidence intervals (CIs). To obtain parsimonious models, all univariate predictors of TTFT (i.e., variables with *p* ≤ 0.05) were tested in Cox analyses with a backward elimination strategy to identify relevant prognostic variables among standard risk factors or miRNAs. Harrell’s C-index, the explained variation in TTFT (an index combining calibration and discrimination), the integrated discrimination improvement (IDI), and the net reclassification index (NRI) were used to assess the accuracy of prognostic models and to measure the gain in prognostic accuracy attributable to relevant miRNAs. To generate a risk prediction rule based on miRNAs, a logistic regression model was fitted with TTFT as the dependent variable and the miRNAs which remained significantly associated with the study outcome by the backward elimination strategy as independent variables. This analysis resulted in an miRNA score (ranging from 0 to 100%), which was then used for further analyses. Statistical calculations were performed using SPSS for Windows v.21 (IBM, Chicago, IL, USA) and Stata 16 (StataCorp, College Station, TX, USA).

## Figures and Tables

**Figure 1 ncrna-10-00046-f001:**
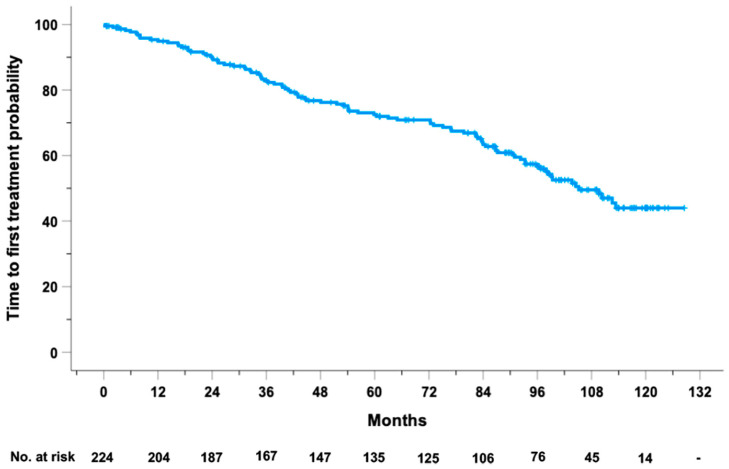
Kaplan–Meier curve of TTFT in the 224 CLL cases from our prospective O-CLL1 study.

**Figure 2 ncrna-10-00046-f002:**
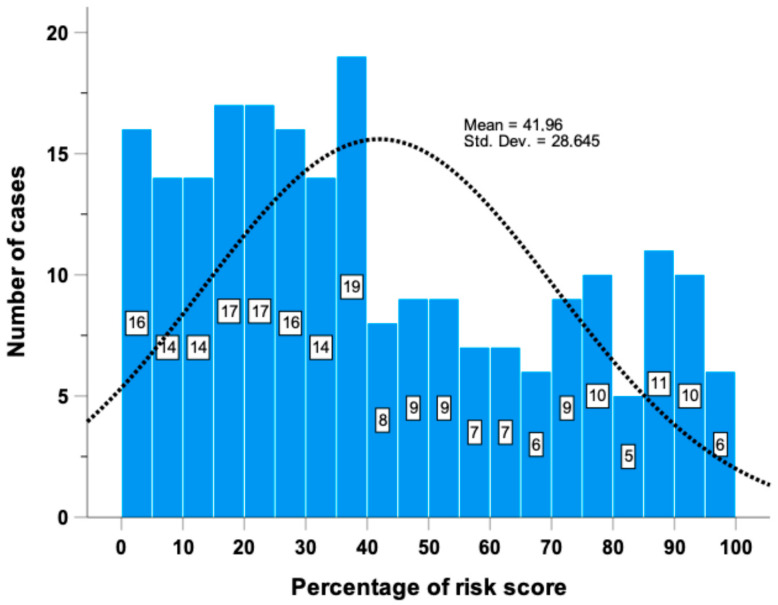
Risk score histogram distribution of 224 cases from O-CLL1 study based on the 16 miRNAs significantly associated with TTFT in multivariate Cox regression model.

**Figure 3 ncrna-10-00046-f003:**
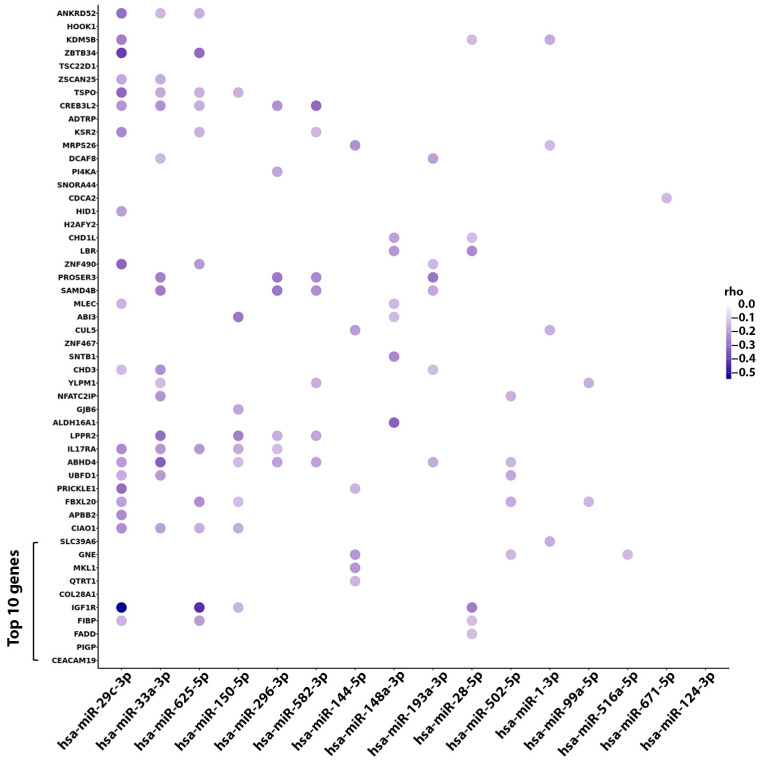
Dot plot displaying the magnitude of negative correlation between the 16 miRNAs significantly associated with TTFT in the 224 CLL cases from our prospective O-CLL1 study and the 50 AI genes from the same cohort. AI genes are sorted by significance magnitude from the bottom to the top. The first 10 genes represent the top AI-based selected genes that were significantly associated with TTFT. The greater intensity of the color in the circle shapes corresponds to a higher magnitude of the negative correlation measured by a Spearman test. rho (or Spearman’s correlation coefficient, measured using a Spearman test) is a non-parametric measure that evaluates the strength and direction of a monotonic relationship between two ordinal or quantitative variables using the ranks of the observations instead of their absolute values.

**Figure 4 ncrna-10-00046-f004:**
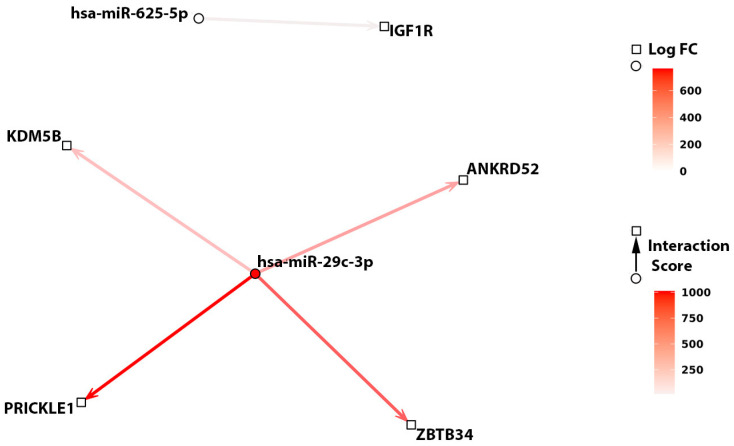
Interaction analysis between miRNAs and genes which were found to be associated with TTFT. Interaction networks between has-miR-29c-3p with 4 AI genes *ANKRD52*, *KDM5B*, *PRICKLE1,* and *ZBTB34* or hsa-miR-625-5p with *IGF1R* are shown. Arrows, circles, and squares indicate the miRNA–mRNA interaction scores, miRNA, and mRNA log-transformed fold-change (Log FC), respectively. Log FC values for miRNA (◯) and mRNA (□) are indicated by a color gradient scale. Regulation score parameter values related to the *IGHV* mutational status (arrows) are highlighted by a scale color gradient from light gray that represents a midpoint value of 0 to red that indicates positive values (interaction score).

**Table 1 ncrna-10-00046-t001:** Cox univariable analyses of miRNAs significantly associated with TTFT.

miRNA-ID	Units of Increase ^1^	HR	95%CI Lower Limit	95% CI Upper Limit	*p*-Value
1-3p	1	1.166	1.006	1.352	0.041
103a-3p	1	0.998	0.996	1.000	0.05
106a-5p	1	1.438	1.003	2.062	0.048
10b-3p	1	1.239	1.055	1.456	0.009
1224-5p	1	1.064	1.024	1.105	0.002
1225-5p	100	1.062	1.001	1.126	0.046
124-3p	1	1.458	1.178	1.805	<0.001
125b-5p	1	0.744	0.56	0.989	0.042
138-5p	1	0.599	0.4	0.896	0.013
140-3p	1	0.991	0.986	0.996	0.001
144-3p	1	1.004	1.002	1.006	0.002
144-5p	1	1.024	1.008	1.041	0.003
146b-5p	1	0.991	0.983	0.998	0.019
148a-3p	1	1.006	1.002	1.010	0.007
150-5p	1000	0.925	0.858	0.997	0.041
150-3p	1	1.035	1.009	1.062	0.008
151-3p	1	0.971	0.946	0.996	0.026
151-5p	1	0.996	0.993	0.998	0.001
155-5p	100	1.058	1.014	1.104	0.009
15a-5p	100	1.074	1.034	1.117	<0.001
184	1	1.472	1.126	1.925	0.005
193a-3p	1	1.132	1.026	1.250	0.014
20a-3p	1	1.049	1.007	1.093	0.022
21-5p	1000	1.152	1.056	1.257	0.001
222-3p	1	0.946	0.907	0.988	0.012
223-5p	1	0.819	0.712	0.943	0.006
24-1-5p	1	1.312	1.013	1.698	0.04
26a-5p	100	0.901	0.813	0.999	0.047
28-5p	1	1.005	1.001	1.009	0.012
296-3p	1	0.570	0.348	0.934	0.026
298	1	1.307	1.022	1.670	0.033
29c-3p	100	0.951	0.923	0.980	<0.001
29c-5p	1	0.952	0.924	0.982	0.002
301a-3p	1	1.038	1.005	1.073	0.024
30c-5p	100	0.621	0.400	0.962	0.033
323-3p	1	0.646	0.426	0.979	0.04
338-5p	1	0.785	0.643	0.960	0.018
339-3p	1	0.689	0.521	0.910	0.009
33a-3p	1	0.375	0.220	0.639	<0.001
361-3p	1	0.989	0.978	0.999	0.034
370	1	1.058	1.028	1.088	<0.001
371-5p	1	1.082	1.016	1.152	0.014
373-5p	1	1.071	1.006	1.14	0.031
376b-3p	1	1.523	1.046	2.218	0.028
491-3p	1	1.766	1.331	2.343	<0.001
500-3p	1	0.817	0.717	0.931	0.002
502-3p	1	0.872	0.796	0.955	0.003
502-5p	1	0.673	0.504	0.898	0.007
513a-5p	1	1.007	1.002	1.012	0.008
518c-5p	1	1.148	1.033	1.276	0.01
520b	1	1.234	1.067	1.426	0.005
532-3p	1	0.898	0.841	0.959	0.001
532-5p	1	0.939	0.891	0.989	0.018
552	1	1.556	1.014	2.388	0.043
557	1	1.197	1.099	1.303	<0.001
566	1	1.616	1.188	2.197	0.002
574-3p	1	1.030	1.006	1.055	0.015
582-3p	1	0.465	0.274	0.789	0.005
584-5p	1	1.162	1.022	1.32	0.022
596	1	0.597	0.391	0.913	0.017
601	1	1.069	1.01	1.131	0.022
603	1	1.552	1.023	2.356	0.039
625-5p	1	0.960	0.940	0.981	<0.001
628-3p	1	0.630	0.417	0.952	0.028
631	1	1.180	1.006	1.385	0.042
645	1	1.604	1.091	2.358	0.016
659-3p	1	1.114	1.01	1.228	0.03
661	1	0.579	0.342	0.981	0.042
665	1	1.145	1.008	1.300	0.037
671-5p	1	1.046	1.014	1.079	0.004
877-5p	1	1.245	1.031	1.503	0.023
9-3p	1	1.086	1.015	1.163	0.017
99a-5p	1	0.615	0.421	0.898	0.012

^1^ To provide a clinically meaningful magnitude of the effect of each miRNA on the study outcome, the units of increase (1, 100, or 1000) were chosen according to the data distribution of the variable.

**Table 2 ncrna-10-00046-t002:** 16 miRNAs independently associated with TTFT after a multivariable Cox regression analysis using a backward elimination strategy.

miRNA-ID	Units of Increase ^1^	HR	95% CI Lower Limit	95% CI Upper Limit	*p*-Value
582-3p	1	0.278	0.145	0.535	<0.001
33a-3p	1	0.334	0.16	0.697	0.003
516a-5p	1	0.490	0.297	0.810	0.005
99a-5p	1	0.512	0.341	0.769	0.001
296-3p	1	0.539	0.301	0.967	0.038
502-5p	1	0.623	0.43	0.905	0.013
625-5p	1	0.958	0.937	0.98	<0.001
29c-3p	100	0.936	0.903	0.970	<0.001
150-5p	1000	1.112	1.005	1.231	0.039
148a-3p	1	1.009	1.004	1.014	<0.001
28-5p	1	1.01	1.005	1.014	<0.001
144-5p	1	1.049	1.026	1.072	<0.001
671-5p	1	1.075	1.027	1.125	0.002
1-3p	1	1.261	1.047	1.517	0.014
193a-3p	1	1.343	1.186	1.52	<0.001
124-3p	1	1.536	1.233	1.913	<0.001

^1^ To provide a clinically meaningful magnitude of the effect of each miRNA on the study outcome, the units of increase (1,100, or 1000) were chosen according to the data distribution of the variable.

**Table 3 ncrna-10-00046-t003:** Prognostic performance of the basic and the expanded models.

	Basic Model	Expanded Model
Harrell’s C-index	75.0%	81.1%
Explained variation in TTFT	45.4%	63.3%
IDI ^1^	-	14.9%, *p* < 0.001
NRI ^2^	-	44.2%, *p* < 0.001

^1^ IDI: integrated discrimination improvement; ^2^ NRI: net reclassification index.

**Table 4 ncrna-10-00046-t004:** Summary of significantly enriched ontologies in the 16 analyzed miRNAs displaying ontology-associated miRNAs and the corresponding adjusted *p*-values and q-values.

Enrichment	ID	Term Specification	Associated miRNA Genes	Adjusted *p*-Value	q-Value
KEGG ORA summary	hsa05206	MicroRNAs in cancer	MIR1-2, MIR28, MIR29C, MIR99A, MIR124-3, MIR150, MIR625	1.41 × 10^−10^	7.41 × 10^−10^
WikiPathways ORA summary	WP299	Nuclear receptors in lipid metabolism and toxicity	MIR33A	0.03	0.009
	WP430	Statin inhibition of cholesterol production	MIR33A	0.03	0.009
	WP1545	miRNAs involved in DNA damage response	MIR29C	0.04	0.01
	WP1601	Fluoropyrimidine activity	MIR29C	0.03	0.009
	WP2023	Cell differentiation expanded index	MIR150	0.04	0.01
	WP2249	Metastatic brain tumor	MIR29C	0.03	0.009

## Data Availability

Datasets used in this study are publicly available and can be accessed from the National Center for Biotechnology Information (NCBI) Gene Expression Omnibus repository (http://www.ncbi.nlm.nih.gov/geo/) accessed on 21 August 2024 and are accessible via GEO Series accession number GSE40533, GSE51529 and GSE40570.

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
