# Peer review of "MicroRNA Profiling as a Predictive Indicator for Time to First Treatment in Chronic Lymphocytic Leukemia: Insights from the O-CLL1 Prospective Study"

_ncrna, 2024, doi:10.3390/ncrna10050046_

Round 1

Reviewer 1 Report

Comments and Suggestions for Authors

Ideally, the improvement of the prognostic prediction models should also be displayed with an AUC or ROC plot, and it would be beneficial to include Kaplan-Meier plots of the 224 cases using their TTFT data for these models.

Figure 3, the figure is difficult to interpret. The authors should explain the meaning of the “rho” in their figure legend. Did the authors identify any seed binding sequences of the 16 miRNAs among these 50 AI genes?

Table 4, the WikiPathways ORA summary result is not particularly meaningful, as only one miRNA is associated with these signaling pathways.

Comments on the Quality of English Language

The language is fine.

Author Response

For the research article ncrna-3149414, “MicroRNA Profiling as a Predictive Indicator for Time to First Treatment in Chronic Lymphocytic Leukemia: Insights from the O-CLL1 Prospective Study.”

Thank you very much for taking the time to review this manuscript. Please find the detailed responses below and the corresponding revisions highlighted in the re-submitted files.

Comment 1: Ideally, the improvement of the prognostic prediction models should also be displayed with an AUC or ROC plot, and it would be beneficial to include Kaplan-Meier plots of the 224 cases using their TTFT data for these models.

Response 1: We acknowledge the reviewer's suggestion to display the improvement of prognostic prediction models using an AUC or ROC plot. However, our study utilized Harrell's C-index as the primary metric for prognostic accuracy. The C-index is analogous to the AUC in ROC analysis but is specifically designed for time-to-event data, an essential consideration that ROC curves do not address. This method accounts for the censored nature of survival data, making it a more appropriate measure for our analysis. Therefore, we chose to employ Harrell's C-index as the indicator of prognostic accuracy in our models.

Additionally, if the reviewer intended to request the inclusion of a survival curve derived from the multivariable Cox proportional hazards model incorporating the risk score, we have addressed this in the revised manuscript. The corresponding survival curve is now presented in Supplementary Figure 1. A visual inspection of this curve reveals a near-identical overlap with the Kaplan-Meier curve previously shown in Figure 1. This concordance suggests that the Cox model, when including the risk score, is well-calibrated for predicting TTFT. Accordingly, we included in the result section the following sentence: "Supplementary Figure 1 illustrates the survival curve derived from the multivariable Cox proportional hazards model, which closely aligns with the Kaplan-Meier curve presented in Figure 1, thereby confirming the robustness and calibration of the model for predicting TTFT."

Comment 2: Figure 3, the figure is difficult to interpret. The authors should explain the meaning of the “rho” in their figure legend. Did the authors identify any seed binding sequences of the 16 miRNAs among these 50 AI genes?

Response 2: As requested by the reviewer we added the meaning of the “rho” in the caption of Figure 3. In addition, we added one supplementary table (Table S2) to show which miRNA-AI gene interactions, with correlation values shown in Figure 3, were experimentally validated (wet-lab or NGS), as reported in Mirtarbase, microCosm_v5, and targetScan_v6.2.

Comment 3: Table 4, the WikiPathways ORA summary result is not particularly meaningful, as only one miRNA is associated with these signaling pathways.

Response 3: Due to their clinical relevance, we focused on the 16 miRNAs significantly associated with TTFT. Although this selection strategy and the limited number of miRNAs may have reduced the likelihood of identifying multiple miRNAs involved in the same biological pathways, it effectively minimized pathway enrichment overlaps. Consequently, this approach decreased FDR associated with multiple comparisons, thereby enhancing the probability of retaining pathways that are supported by a single miRNA. Nevertheless, we decided to present the Wikipathway ORA summary annotation to more comprehensively describe the functional roles of these 16 miRNAs.

Reviewer 2 Report

Comments and Suggestions for Authors

In the research article titled “MicroRNA Profiling as a Predictive Indicator for Time to First 2 Treatment in Chronic Lymphocytic Leukemia: Insights from 3 the O-CLL1 Prospective Study” the authors studied the correlation of 513 microRNAs (miRNAs) to the time to first treatment in 224 patients with CLL. The study found 73 miRNA expression associated with TTFT in the CLL patients. Furthermore, the authors also found 16 miRNAs retained an independent relationship with the outcome in a multivariable analysis. The article is written very well and the authors have made consideration of recent advancements in the field of miRNA-based biomarker discovery. This article is good for publication in the journal with a minor inclusion of analysis  which might improve the study outcome,

1.      Authors have mentioned that incorporating their previously discovered AI gene expression data in the foundational model led to an enhanced model predicting TTFT. It would be great if authors can incorporate the miRNA based predictive model in the foundational model to see the improvement in predicting TTFT and the significance of prognostic efficacy.

Author Response

For the research article ncrna-3149414 “MicroRNA Profiling as a Predictive Indicator for Time to First Treatment in Chronic Lymphocytic Leukemia: Insights from the O-CLL1 Prospective Study.”

Thank you very much for taking the time to review this manuscript. Please find the detailed responses below and the corresponding revisions highlighted in the re-submitted files.

In the research article titled “MicroRNA Profiling as a Predictive Indicator for Time to First Treatment in Chronic Lymphocytic Leukemia: Insights from the O-CLL1 Prospective Study” the authors studied the correlation of 513 microRNAs (miRNAs) to the time to first treatment in 224 patients with CLL. The study found 73 miRNA expression associated with TTFT in the CLL patients. Furthermore, the authors also found 16 miRNAs retained an independent relationship with the outcome in a multivariable analysis. The article is written very well and the authors have made consideration of recent advancements in the field of miRNA-based biomarker discovery. This article is good for publication in the journal with a minor inclusion of analysis which might improve the study outcome,

Comment: Authors have mentioned that incorporating their previously discovered AI gene expression data in the foundational model led to an enhanced model predicting TTFT. It would be great if authors can incorporate the miRNA based predictive model in the foundational model to see the improvement in predicting TTFT and the significance of prognostic efficacy.

Response: In response to the reviewer's suggestion, we have conducted the analysis integrating the miRNA-based predictive score into the foundational model. Our results indicate that the miRNA-based risk score achieved a discriminatory power of 79% in predicting TTFT. However, we would prefer to clarify that the main goal of the analysis is not just to show how accurate the new model is by itself, but to assess whether it provides additional valuable prognostic information to a pre-existing model that already accounts for known and easy-to-detect risk factors. This distinction is important in scientific research, as it focuses on the practical utility of the new model in improving predictions beyond what is already known, rather than simply evaluating its standalone performance. For this reason, we prefer to retain the emphasis as presented in our original analysis, as it better aligns with our objective of evaluating the added value of the risk score within the broader predictive context.